# The Development of a Multi-Modality Emotion Recognition Test Presented via a Mobile Application

**DOI:** 10.3390/brainsci12020251

**Published:** 2022-02-11

**Authors:** Rwei-Ling Yu, Shu-Fai Poon, Hsin-Jou Yi, Chia-Yi Chien, Pei-Hsuan Hsu

**Affiliations:** Institute of Behavioral Medicine, College of Medicine, National Cheng Kung University, Tainan 701, Taiwan; s86095021@gs.ncku.edu.tw (S.-F.P.); yiicindy27@gmail.com (H.-J.Y.); a0918732736@gmail.com (C.-Y.C.); kusano0704@gmail.com (P.-H.H.)

**Keywords:** emotion recognition, performance-based analysis, multi-modalities, mobile application

## Abstract

Emotion recognition ability is the basis of interpersonal communication and detection of brain alterations. Existing tools for assessing emotion recognition ability are mostly single modality, paper-and-pencil test format, and using only Western stimuli. However, various modalities and cultural factors greatly influence emotion recognition ability. We aimed to develop a multi-modality emotion recognition mobile application (MMER app). A total of 169 healthy adults were recruited as participants. The MMER app’s materials were extracted from a published database, and tablets were used as the interface. The Rasch, factor analysis, and related psychometric analyses were performed. The Cronbach alpha was 0.94, and the test–retest reliability was 0.85. Factor analyses identified three factors. In addition, an adjusted score formula was provided for clinical use. The MMER app has good psychometric properties, and its further possible applications and investigations are discussed.

## 1. Introduction

### 1.1. Emotion Recognition

Emotion recognition indicates the identification of emotional information in various ways, including through the face and prosody [1]. Different receiving channels affect emotion recognition accuracy. Emotion recognition ability is essential for human social interaction and considerably impacts everyday life [2]. When individuals interact with others, they need to receive and recognize emotional information from various sources, such as verbal (e.g., words and semantics) or non-verbal (e.g., tones, facial emotions, and body movements) stimulation. Studies have shown that emotion recognition ability is an essential factor in developing the ability to “understand others’ intentions” [3]. In the past few years, research on emotion recognition has grown tremendously and attracted the interest of scholars in various fields.

Emotion recognition ability develops during childhood [4] and changes with age. Nevertheless, the relationship between emotion recognition ability and aging is inconclusive. Previous studies have shown that emotion recognition ability declines with age [5]. The elderly find it more challenging to identify anger [6], fear, and sadness [7] than young people. Moreover, emotion recognition is essential for detecting brain alterations or early signs of psychological/neurological disorders. Evidence has shown that patients with neurodegenerative diseases (e.g., Alzheimer’s disease [8] and Parkinson’s disease [9]) or psychiatric disorders (e.g., autism spectrum disorder and schizophrenia [10]) are found to have impairments in facial emotion recognition ability. Furthermore, it has also been shown that alexithymia is a transdiagnostic construct [11], and detecting difficulties in emotion recognition could facilitate the diagnosis of psychiatric disorders. Changes in emotion recognition ability may reflect alterations in the corresponding brain areas. The literature indicates that some brain areas (e.g., the striatum, fusiform gyrus, superior temporal gyrus, amygdala, orbitofrontal cortex, basal ganglia, somatosensory, and insula) might be responsible for this ability [12,13,14]. 

Based on empirical evidence, it is believed that emotion recognition ability may be a good indicator of detecting subtle changes in the brain and facilitating the diagnosis and management of psychiatry-related diseases. Therefore, an assessment tool that measures emotion recognition ability is essential and has clinical utility.

### 1.2. Measurement of the Emotion Recognition

#### 1.2.1. Single-Modal Emotion Recognition Tests

Most previous recognition tests have been based on Ekman’s facial emotion database [15,16]; however, most current emotion recognition tests only focus on four basic emotions (i.e., happy, sad, angry, and surprised/frightened) [17]. Unfortunately, only a few Eastern facial stimuli were included in this database. Although facial expression is believed to be a universal language for emotion [18], studies have found that ethnicity and cultural factors must be considered [19,20]. Culture may affect how individuals express their feelings; Westerners use the whole face to express emotions, while Easterners use the upper half of the face [20]. Jack and colleagues (2012) [21] found that Asians have difficulty recognizing European facial emotions. Current studies have revealed a culture-specific decoding strategy applied in various ethnicities and cultures [20].

Few single-modal emotion recognition tests (shown in Table 1), which were applied Eastern materials, have been developed in the past, including the Japanese and Caucasian Brief Affect Recognition Test (JACBART) [22] and the Chinese Facial Emotion Recognition Database (CFERD) [23]. The JACBART rates the degree in the seven types of facial emotions simultaneously (i.e., anger, contempt, disgust, fear, happiness, sadness, and surprise) for one facial stimulus. The CFERD is a three-dimensional color facial emotion recognition task, which involves matching each facial expression with the corresponding emotional word (happiness, disgust, fear, anger, sadness, surprise, or neutral). The tests mentioned above address the culture-specific issues but do not consider the interaction between the facial emotion source and other modalities. Therefore, developing a multi-modal emotion recognition test with Eastern materials is needed.

#### 1.2.2. Multi-Modal Emotion Recognition Tests

A few emotion recognition tests have examined dual or multi-modal emotion recognition (shown in Table 2). The most widely known are the Florida Affect Battery (FAB) [24] and the Diagnostic Analysis Nonverbal Accuracy Scale (DANVA) [17]. These tests are well known and commonly used dual emotional recognition tests; however, they have limitations. For instance, the lack of core emotions (i.e., the FAB lacks disgust and fear, and the DANVA lacks disgust and surprise), the application of only Caucasian faces as stimuli, and the use of only the female face and voice. Fear and disgust are crucial emotions for people’s survival and social interactions. If one cannot recognize another’s fear, it may affect the individual’s understanding of a crisis and messages from others. It is believed that disgust is related to human beings’ specific needs (e.g., hunger and moral) [25], and disgust allows individuals to perform correct behaviors, such as nausea and vomiting after eating expired food; in addition, sexual violence can cause human disgust. Thus, the recognition of these two emotions is crucial for an individual’s life and interpersonal relationships. Although the FAB has been translated and validated in the Chinese population [26], it still uses Caucasian faces for facial stimuli. The FAB altered the voice to Chinese pronunciation; however, the Caucasian faces with Chinese voices may confuse the participants. 

The Awareness of Social Inference Test Emotion Evaluation Test [27], the Multimodal Emotion Recognition Test (MERT) [28], the Geneva Emotion Recognition Test (GERT) and its short form (GERT-S) [29,30], and the Emotion Recognition Assessment in Multiple Modalities test [31] were also developed for examining multi-modal emotion recognition; however, they were not similar to the FAB with multiple subtests and were only performed for matching tests, involving different types of stimuli and emotional words. Moreover, most of these tests examine more than ten types of emotions, and too many emotion choices may interfere with recognition. The stimuli for the above multi-modal emotion recognition tests were all Caucasian faces, and they were not validated for the Eastern population.

#### 1.2.3. The Unmet Need for, and Challenges of, the Development of the Emotion Recognition Test

The development of emotion recognition tools has progressed from the single-channel format [22,23,32,33] to the exploration of emotion recognition ability in multiple channels [17,24,26,27,28,29,30,31]. Besides, rather than focusing on basic emotions in the past, a more comprehensive range of emotions and covering various emotions is needed for recent development tools [28,29,30,31]. Another unmet need is that most tools use stimulus material from Western countries or ethnicities, and few tests have been developed for Eastern populations. Furthermore, traditional paper-and-pencil tests have practical limitations. Through using readily available electronic tools as a medium, it will be possible to enhance the accessibility and applicability of the tools. However, to the best of our knowledge, none of the tools for assessing emotion recognition ability in the literature reviewed above were displayed through user-friendly methods (e.g., mobile application).

**Table 1 brainsci-12-00251-t001:** Single-modality emotion recognition tests with eastern facial stimulus.

Test Name	Country	Number of Items	Test Methods or Procedures	Reliability Validity
Japanese and Caucasian Brief Affect Recognition Test (JACBART) [22]	Japan	56 ^a^	Rate the degree of the 7 emotions	Internal reliability Test–retest Reliability Concurrent validity Convergent validity
The Chinese Facial Emotion Recognition Database (CFERD) [23]	Taiwan	100 ^a^	7 emotions classification	NA
NA [32]	Malaysia	56 ^a^	7 emotions classification	NA
Chinese Affective Picture System (CAPS) [33]	China	60 ^a^	4 emotions classification	NA

NA—not available. Type of the items: ^a^—images.

**Table 2 brainsci-12-00251-t002:** Multi-modality emotion recognition tests with Western facial stimulus.

Test Name	Country	Number of Items	Test Content	Test Methods or Procedures	Reliability Validity
Florida Affect Battery [24]	US	232 ^a,b,c^	Similar to MMER	5 emotions classification	Test–retest 0.89–0.97
The Awareness of Social Inference Test (TASIT)—part 1: Emotion Evaluation Test (EET) [27]	Australia	28 ^e^	Similar to subtest 3	7 emotions classification	Parallel forms Reliability Construct validity
Multimodal Emotion Recognition Test (MERT) [28]	Switzerland	120 ^a,b,d,e^	Similar to subtest 3	10 emotions classification	Interrater 0.38 Test–retest 0.78 EFA
Florida Affect Battery (Chinese version) [26]	Taiwan	225 ^a,b,c^ (Western facial stimulus; Chinese prosody stimulus)	Similar to MMER	5 emotions classification	Content validity Criterion validity Norm comparison
Geneva Emotion Recognition Test (GERT) [29]	Switzerland	108 ^e^	Similar to subtest 3	14 emotions classification	NA
Geneva Emotion Recognition Test—short form (GERT-S) [30]	Switzerland	42 ^e^	Similar to subtest 3	14 emotions classification	Cronbach alpha = 0.80 CFA good fit
Emotion Recognition Assessment in Multiple Modalities Test (ERAM) [31]	Sweden	72 ^a,b,c^	Similar to subtest 3	12 emotions classification	Cronbach alpha = 0.74 CFA good fit

NA—not available. The types of the items: ^a^—images; ^b^—audio; ^c^—audio–image; ^d^—video; ^e^—audio–video.

### 1.3. Aim

We aimed to develop the Multi-Modality Emotion Recognition mobile application (MMER app), a multi-modal emotion recognition test. Chinese faces and prosody were used as stimuli. Moreover, comprehensive emotions (i.e., neutral, happy, sad, angry, disgust, fear, and surprise) were included. Furthermore, our test was designed as a device application and performed on a tablet due to the popularity and handiness of mobile devices. We used the tablet as an interface to calculate and output the scores, which improves the convenience and applicability of the test.

## 2. Materials and Methods

### 2.1. Participants

We recruited 169 healthy adults (demographic characteristics are presented in Table 3) from community activity centers and our college. Participants suspected to have dementia, based on a mini mental state examination (MMSE) score below 24, were excluded from the study [34]. In addition, patients with a history of psychiatric illness, substance abuse, severe systemic diseases, and traumatic brain injury were excluded. Before participation, the participants provided informed consent; ethical standards were drawn up based on the 1964 Declaration of Helsinki. The ethical research committee of National Cheng Kung University Hospital IRB (approval number: A-ER-107-425) confirmed the study protocols.

### 2.2. Procedure of the Development of the MMER App

#### 2.2.1. Item Generation of the MMER App 

We generated the items and format of the MMER app through a literature review and reference to other tests (e.g., the FAB). Facial and prosodic emotions were chosen as the materials for the MMER app. The materials were obtained from the Emotional Speech Database in Taiwan and from the Taiwan Corpora of Chinese Emotion and Relevant Psychophysiological Database [35,36,37]. Those datasets included many Eastern faces and tones. We randomly selected 2272 Chinese face pictures and 188 sound segment stimuli in 7 emotions to establish the MMER app. The pretest version of the MMER app included 9 subtests (5 facial tests, 2 prosodic tests, and 2 cross-modal tests), with a total of 325 items. 

#### 2.2.2. The Subtests of the MMER App

We referred to the format of the FAB to generate the subtests. Subtest 1 was a facial feature discrimination test. Subtests 2–5 were facial-related emotion recognition tests, subtests 6 and 7 were prosodic-related emotion recognition tests, and subtests 8 and 9 were facial–prosodic, cross-modal emotion recognition tests. We wrote the items into the app and used the tablet to collect the participants’ responses.

Subtest 1 was a “facial feature discrimination test”, with 24 items, including the front, side, and two-thirds of the face. Two faces were presented in the tablet at a time, and the participants were asked to determine whether the two faces were the same person. Subtest 2 was a “facial emotion discrimination test” using faces. The subtest had 35 items. Two faces were presented in the tablet at a time, and the participants were asked to determine whether the emotions of the two faces were the same. Subtest 3 was a “face-word matched test” and contained 42 items, including front, side, and two-thirds of the face. One face and seven emotion terms (i.e., neutral, happy, sad, angry, disgust, fear, and surprise) were displayed on the tablet, and the participants were asked to choose one emotion term that best fit the target face. Subtest 4 was a “word-face matched test” and contained 14 items. One emotion term and seven faces were shown on the tablet, and the participants were asked to choose one face that best fit the target emotion term. Subtest 5 was a “face-face emotion mated test” containing 28 items. Six faces were shown in the tablet (along with the target face), and the participants were asked to select one face whose emotion best resembled that of the target face.

Subtest 6 was a “prosodic emotion discrimination test”. Two emotional sentences were played via the tablet, and the participants were asked to determine whether the two sentences displayed the same emotions. Subtest 7 was a “prosodic-word matched test” with 28 items. One sentence and seven emotion terms were presented, and participants were asked to choose one emotion term that most suitably represented the target sentence’s emotion.

Subtest 8 was a “prosodic-face matched test” and contained 35 items. One sentence and four faces were presented on the tablet, and participants were asked to select one face whose emotion was the most suitable for the target sentence’s emotion. Subtest 9 was a “face-prosodic matched test” and contained 42 items. One face and four sentences were presented, and participants were asked to select one sentence whose emotion was the most suitable for the target face’s emotion.

#### 2.2.3. The Pretest Stage

Seven participants were recruited to join the pretest procedure to examine the reaction time, test item accuracy, and modification of some items. In addition, we collected the participants’ comments after completing the test to improve the quality of the MMER app. After the pretest, we kept the items with an accuracy above 50%, increased the number of practice items (subtests 3–5 and subtest 7), and added feedback in the practice section. We modified the items in which accuracy was below 50% by excluding stimuli that presented conflicting emotions (e.g., actors acting out sad emotions, but most participants rated it as scared). The option of subtest 9 was originally a four-segment prosodic emotion; however, considering that prosodic emotion has only two opportunities to play and is more likely to be affected by the participant’s cognitive function (e.g., poor memory may make them forget the sound of the previous option), the number of options were reduced to three. In the analysis of the previous version of the MMER app, subtest 2 had unacceptable internal consistency (Cronbach’s alpha = 0.46) and test–retest reliability (0.28). Subtest 6 had poor internal consistency (Cronbach’s alpha = 0.56) and questionable test–retest reliability (0.67). Thus, we deleted these two subtests. The Rasch analysis results were used to modify the MMER app. Based on the Rasch analysis, the comparison between participant ability and item difficulty was conducted, and the items below participant ability were deleted.

Item difficulty, which was below the minimum value of participant performance, was eliminated. Nevertheless, it retained half of the items in each emotion if over half of the items in each emotion were canceled. As the purpose of the MMER app was to determine emotion recognition ability, seven types of emotion items were needed to fulfill its aims, even though it was easy for most of the population. The retained items were selected based on difficulty—the higher difficulty items were retained until each emotion item reached half.

#### 2.2.4. The MMER App

The MMER app included 7 subtests and 198 items (100 items of facial tests, 25 items of prosodic tests, and 73 items of cross-modal tests), with a scoring method of 1 point per question (total score of 198 points).

### 2.3. Measurement

The MMER app was displayed on a 10.1-inch tablet to evaluate the participants’ emotion recognition ability (Figure 1). We used the MMSE [34] to exclude participants with dementia. MMSE has a total score of 30 and is often used to assess an individual’s general cognitive function. Those with scores below 24 were considered likely to have dementia. The Reading Mind in the Eyes Test (RMET) [38] was applied to establish criterion-related validity. The RMET is a test that is often used to assess individuals’ judgment of others’ feelings or emotions through the expressions around their eyes. This test presents a series of black-and-white photos expressing emotions and asks participants to choose the adjective that best fits the emotion expressed by the people in the photos.

### 2.4. Data Analysis

Descriptive statistics of performance were calculated for total scores, accuracy in each subtest, and demographic characteristics of all participants. Cronbach’s alpha was used to calculate internal consistency to examine the internal reliability of the MMER app. Twenty-eight participants were invited to complete the test again after three–four months from the first visit, and Pearson’s correlation was employed to confirm the test–retest reliability. The RMET [37] was applied to establish criterion-related validity. Confirmatory factor analysis was conducted to examine the factorial structure of the MMER app. A one-factor model (model 1) was first used, as the purpose of the MMER app was related to one factor—“emotion recognition”. Multi-dimensional models were conducted to examine the relationship between factors; models 2 and 4 were oblique, and models 3 and 5 were orthogonal. Moreover, two-factor models (models 2 and 3) and three-factor models were also tested to study the relationship between the three theoretical concepts (facial recognition, facial emotion recognition, and prosody emotion recognition). Oblique and orthogonal is a factor rotation for transform gained factors from factor analysis. It would maximize the large factor loadings and minimize the small factor loadings to enhance the interpretability for the factors. The major difference between oblique and orthogonal rotation is that the factors in the oblique rotation model could be correlated, and the correlation between factors in the orthogonal rotation model is equal to zero.

## 3. Results

### 3.1. Performance

The mean and standard deviation of the demographic characteristics and performance of the participants are presented in Table 3. The correct score and accuracy of the MMER app for the seven types of emotion are shown in Table 4.

### 3.2. Reliability, Confirmatory Factor Analyses, and Criterion-Related Validity

#### 3.2.1. Reliability

The internal consistency of the MMER app was excellent (Cronbach’s alpha = 0.94) with good test–retest reliability (0.87).

#### 3.2.2. Confirmatory Factor Analysis

The results for the different models are presented in Table 5, Figure 2, Figure 3 and Figure 4. The fit index showed that the orthogonal models fit more with the data. The results of Models 2 and 4 are similar. Moreover, the structure of Model 4 is equal to the theoretical structure of the MMER app. The factor loadings of Model 4 are shown in Table 6.

#### 3.2.3. Criterion-Related Validity

The result of the Pearson’s correlation indicated that there was a strong positive association between RMET and the total score of the MMER app (r = 0.53, *p* < 0.001). Moreover, all subtests were also significantly correlated with RMET. For subtest 3 (“face-word matched test”), the Pearson correlation showed a strong positive association with RMET (r = 0.54, *p* < 0.001). In addition, the Pearson correlation showed that the RMET had a moderate positive association with subtest 4 (“word-face matched test”) (r = 0.43, *p* < 0.001), subtest 5 (“face-face emotion mated test”) (r = 0.47, *p* < 0.001), subtest 7 (“prosodic-word matched test”) (r = 0.40, *p* < 0.001), subtest 8 (“prosodic-face matched test”) (r = 0.43, *p* < 0.001), and subtest 9 (“face-prosodic matched test”) (r = 0.41, *p* < 0.001). For subtest 1 (“facial feature discrimination test”), the Pearson correlation showed a weak positive association with RMET (r = 0.23, *p* < 0.001).

### 3.3. Multiple Stepwise Regression for Application

According to the correlation analysis, the demographic variables of the sample were correlated with the MMER app; for instance, age (r = −0.77, *p* < 0.001) and education (r = 0.41, *p* < 0.001) were correlated to the total score. Multiple stepwise backward regression was conducted to modify the score of the MMER app using a formula. The demographic variables, including age, gender, and education, explained 65.67% of the MMER app performance based on multiple stepwise backward regression. The adjustment formula was as follows:Adjusted MMER app score =MMER app total score-4.007(gender-0.33)-0.750(age-48.28) + 1.645(education-13.73)where gender was defined as male = 1 and female = 0.

## 4. Discussion

Emotion recognition is vital for social interaction and intimacy, and the evaluation of emotion recognition ability is necessary for detecting brain dysfunction and further developing rehabilitation programs. In the current study, we have overcome the limitations of previous studies and use the tablet’s advantages to develop a reliable and effective emotion recognition test—the MMER app. The MMER app contains 7 subtests with 198 items. The stimuli used in the MMER app are all Eastern stimuli (both sexes), and the MMER app can measure multi-modal emotion recognition ability (i.e., face and prosody). The MMER app has good psychometric properties and takes only 20 minutes to complete. To the best of our knowledge, the MMER app is the first suitable test in the Chinese population to measure emotion recognition ability via various modalities (i.e., visual and auditory).

The total accuracy of the MMER app was 69%, and subtest 1 had the highest accuracy (93%). Compared with other subtests, healthy people generally have the highest accuracy in subtest 1. The face-related subtest (76%) had the second-highest accuracy, followed by the face–prosody subtest (61%). The prosody-related subtest (44%) had the lowest accuracy. The accuracy was similar to previous study findings, which reported facial emotion recognition accuracy between 65 and 78% and prosodic emotion recognition between 52 and 65% [39,40]. The highest accuracy of each emotion was anger, followed by happiness; the lowest accuracy was fear, followed by disgust. These findings are consistent with previous research [35,37] and indicate that people have the highest consistency in evaluating happiness and anger and the lowest consistency in assessing fear and disgust. The highest accuracies of the face-related and prosodic-related subtests were for happiness and anger, respectively. In addition, the lowest accuracy of the face-related subtest was for fear. For the prosody-related subtest, the lowest accuracy was for disgust, and these findings are consistent with Scherer et al.’s results. [40].

The MMER app has high internal consistency and adequate test–retest reliability. Moreover, the RMET, as criterion-related validity, is closely related to the MMER app. Among the subtests, the face-related subtests and the RMET scores, which also use facial pictures as stimulus materials, are significantly correlated. Previous studies have shown that these two abilities are highly related [41], and lesion studies have also found that emotion recognition and mind-reading abilities partially share the exact neural mechanisms [13]. Thus, we believe that the MMER app has well-established, criterion-related validity. In addition, the factor analyses confirm that the MMER app has three factors and concepts: facial recognition and facial and prosody emotion recognition. The findings of our factor analyses show that the MMER app is a multi-modality tool for measuring emotion recognition ability. Moreover, the seven subtests may represent a different psychological mechanism of emotion recognition owing to the different presentation methods. Further investigations that apply this test in clinical practice—focusing on the relationship between different types of emotion recognition defects and related brain pathologies, which may assist in disease detection and rehabilitation—are encouraged.

The MMER app has a certain degree of discrimination on healthy participants, which is different from previous tests (e.g., FAB and DANVA) that have a ceiling effect. Studies suggest that demographic variables, such as age, gender [42], and education level, are crucial in emotion recognition ability [43]. Multiple stepwise regression can be adjusted due to demographic variables, potentially being used as the norm exploration in the future. An individual’s emotion recognition ability can be determined after using the adjusted score to query the percentage level comparison table. If one’s adjusted MMER app score is below the common standard (5th percentile), his/her emotion recognition ability is considered defective. This is the first multi-modality emotional recognition test for an app to the best of our knowledge. Our MMER app can improve the accessibility and clinical efficiency of assessment. 

The limitation of this study lies in the lack of developed Eastern emotion recognition models to compare with and refer to, which is also the reason for this study. Although the MMER app is significantly related to RMET, and criterion-related validity was established, the RMET cannot perfectly play the role of the criterion, especially the part of voice–emotion subtests. Second, the cultural divergence among Eastern countries is considerable. Third, the intensity of emotional stimuli was not considered in this study. Emotional intensity may be one reason that affects the ability to recognize emotions and needs further investigation. Fourth, the accuracy of the prosody-related subtests was low. Previous studies have found that other factors easily affect prosody-related emotions (e.g., semantic meaning) [31]. Our findings also confirmed that prosody-related emotion recognition accuracy is lower than face-related emotion recognition. Although the accuracy of these subtests was poor, other psychometric properties of these subtests were acceptable–good. Finally, only healthy participants were recruited because there is evidence that psychiatric disorders may impair emotion recognition. The MMER app could potentially be created as a screening test or a standard emotion recognition test within Eastern cultures. Thus, other populations should be recruited in future studies to validate our findings. Modification of the MMER app was also required for a prospective study on sample variation, multicultural comparison in Eastern culture, and item selection. 

## 5. Conclusions

The MMER app has well-established psychometric properties and provides an integrated Eastern version of an emotion recognition test, with multiple modalities involved in comprehensive emotions, and no sex bias in the stimulus. Moreover, we offered a formula to generate the adjusted score, which can be used to determine whether an individual’s emotion recognition ability is impaired through the percentile scale correspondence. Further research is needed to recruit other populations (e.g., clinical cases) to cross-validate the MMER app. This test also has the potential to be used in future clinical practice. 

## Figures and Tables

**Figure 1 brainsci-12-00251-f001:**
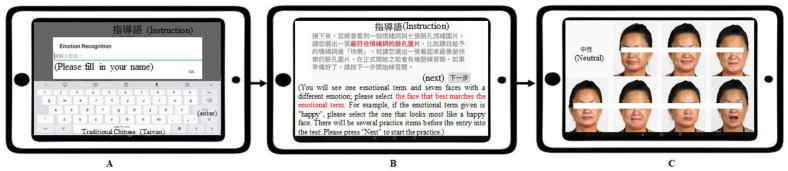
The diagrammatic sketch of the MMER app, taking subtest four as an example. (**A**) The step in which the participant’s information was entered; (**B**) the step to show the participants the instructions; (**C**) the step where the participant was asked to choose the correct answer.

**Figure 2 brainsci-12-00251-f002:**
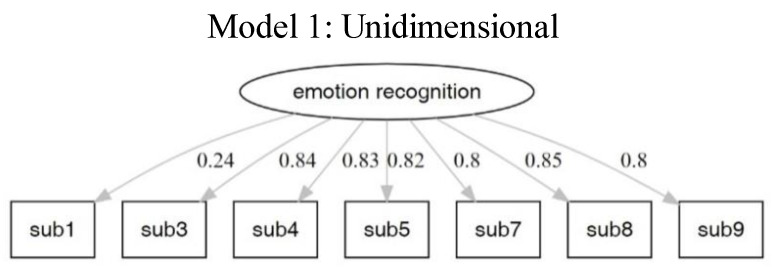
Model 1 is a one-factor (emotion recognition) model.

**Figure 3 brainsci-12-00251-f003:**
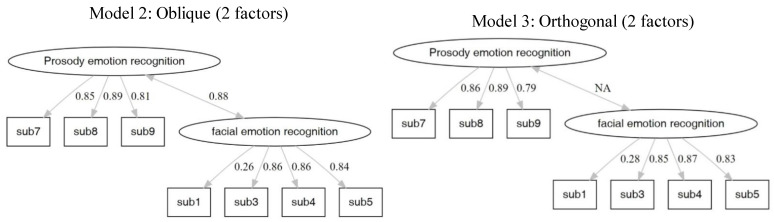
Model 2 and 3 are two-factor (prosody emotion recognition and facial emotion recognition) models with oblique and orthogonal rotation, respectively.

**Figure 4 brainsci-12-00251-f004:**
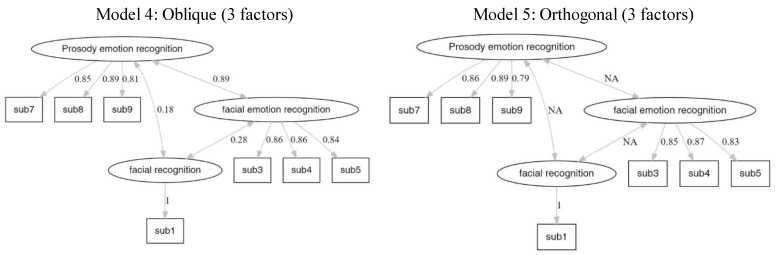
Model 4 and 5 are three-factor (facial recognition, prosody emotion recognition, and facial emotion recognition) models with oblique and orthogonal rotation, respectively.

**Table 3 brainsci-12-00251-t003:** The demographic characteristics and performance of participants in the MMER app.

**Demographic Variables**	**Mean**		**Range**	
Gender (male %)	32.54 (46.99%)		-	
Age, years	48.28 (20.27 ^+^)		18–80	
Education, years	13.73 (2.97 ^+^)		6–20	
**Subtests of the MMER App**	**Full Mark**	**Correct Score** **Mean (SD)**	**Correct Score Range**	**Accuracy** **Mean (SD)**
Total score	198	135.88 (22.14)	71.00–174.00	0.68 (0.11)
Subtest 1	24	22.42 (2.09)	13.00–24.00	0.93 (0.09)
Subtest 3	30	22.14 (4.08)	10.00–29.00	0.74 (0.14)
Subtest 4	20	15.74 (2.82)	6.00–20.00	0.79 (0.14)
Subtest 5	26	20.07 (3.96)	3.00–26.00	0.77 (0.15)
Face-related subtests ^a^	100	57.95 (9.85)	24.00–74.00	0.76 (0.13)
Subtest 7 (prosody-related subtest)	25	11.12 (3.90)	3.00–21.00	0.44 (0.16)
Subtest 8	35	20.62 (5.19)	5.00–30.00	0.59 (0.15)
Subtest 9	38	23.76 (5.25)	8.00–35.00	0.63 (0.14)
Face–prosody subtest ^b^	73	44.38 (9.63)	22.00–65.00	0.61 (0.13)

Abbreviations: MMER app—Multi-Modalities Emotion Recognition Mobile Application; SD—standard deviation; ^a^—sum of subtests 1, 3, 4, and 5; ^b^—sum of subtests 8 and 9; ^+^—standard deviation.

**Table 4 brainsci-12-00251-t004:** Correct score and accuracy of the MMER app in 7 types of emotion.

Emotion	Full Mark	Correct Score Mean (SD)	Correct Score Range	Accuracy Mean (SD)
Total score	174	113.46 (21.58)	49.00–151.00	0.66 (0.12)
Neutral	25	17.20 (4.12)	4.00–25.00	0.69 (0.16)
Happiness	21	15.89 (2.61)	9.00–21.00	0.76 (0.12)
Sadness	23	15.91 (4.02)	4.00–22.00	0.69 (0.17)
Angry	25	20.17 (3.42)	9.00–25.00	0.84 (0.14)
Disgust	29	15.78 (4.43)	4.00–24.00	0.54 (0.15)
Fear	29	13.51 (4.60)	4.00–23.00	0.47 (0.16)
Surprise	22	15.01 (3.43)	3.00–21.00	0.68 (0.16)
Version and subtests of the MMER app				
Face-related subtests ^a^	76			
Neutral	10	7.88 (1.69)	2.00–10.00	0.79 (0.17)
Happiness	8	7.67 (0.72)	3.00–8.00	0.96 (0.09)
Sadness	10	7.33 (1.75)	2.00–10.00	0.73 (0.17)
Angry	12	10.47 (1.62)	4.00–12.00	0.87 (0.13)
Disgust	14	10.76 (2.98)	1.00–14.00	0.77 (0.21)
Fear	14	6.90 (3.05)	1.00–14.00	0.49 (0.22)
Surprise	8	6.95 (1.21)	1.00–8.00	0.87 (0.15)
Prosody-related subtest (subtest 7)	25			
Neutral	4	2.27 (1.17)	0.00–4.00	0.57 (0.29)
Happiness	3	1.33 (0.92)	0.00–3.00	0.44 (0.31)
Sadness	4	2.21 (1.23)	0.00–4.00	0.55 (0.31)
Angry	2	1.59 (0.59)	0.00–2.00	0.80 (0.30)
Disgust	4	0.93 (0.92)	0.00–3.00	0.23 (0.23)
Fear	4	1.33 (1.03)	0.00–4.00	0.33 (0.26)
Surprise	4	1.46 (0.97)	0.00–4.00	0.37 (0.24)
Face–prosody subtest ^b^	73			
Neutral	11	7.05 (2.43)	1.00–11.00	0.64 (0.22)
Happiness	10	6.88 (1.83)	2.00–10.00	0.69 (0.18)
Sadness	9	6.37 (1.88)	1.00–9.00	0.71 (0.21)
Angry	11	8.11 (2.12)	1.00–11.00	0.74 (0.19)
Disgust	11	4.08 (1.87)	0.00–9.00	0.37 (0.17)
Fear	11	5.29 (1.79)	1.00–9.00	0.48 (0.16)
Surprise	10	6.60 (2.09)	1.00–10.00	0.66 (0.21)

Abbreviations: please see Table 3. ^a^—sum of subtests 3, 4, and 5; ^b^—sum of subtests 8 and 9.

**Table 5 brainsci-12-00251-t005:** Confirmatory factor analysis of MMER app with different models.

Model	χ^2^	df	RMSEA (90% CI)	CFI	TLI	AIC
Model 1: Unidimensional	45.282	14	0.115 (0.079–0.153)	0.958	0.936	7425.075
Model 2: Oblique (2 factors)	12.444	13	<0.001 (0.000–0.074)	1.000	1.001	7394.237
Model 3: Orthogonal (2 factors)	174.631	14	0.261 (0.227–0.296)	0.782	0.673	7554.424
Model 4: Oblique (3 factors)	10.787	12	<0.001 (0.000–0.072)	1.000	1.003	7394.580
Model 5: Orthogonal (3 factors)	186.673	15	0.260 (0.228–0.294)	0.767	0.674	7564.466

χ^2^—chi-square; df—degree of freedom; RMSEA—root mean square effort of approximation; CFI—comparative fit index; TLI—Tucker–Lewis Index; AIC—Akaike.

**Table 6 brainsci-12-00251-t006:** Factor loadings for confirmatory factor analysis with models 4.

Subtests	Factors		
	Facial Recognition	Facial Emotion Recognition	Prosody Emotion Recognition
1	1.000		
3		0.861	
4		0.858	
5		0.840	
7			0.846
8			0.891
9			0.807

## Data Availability

The dataset can be obtained with the corresponding author’s permission.

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
