# Peer review of "The Development of a Multi-Modality Emotion Recognition Test Presented via a Mobile Application"

_brainsci, 2022, doi:10.3390/brainsci12020251_

Round 1

Reviewer 1 Report

In this work, the authors present the design of the Multi-Modality Emotion Recognition (MMER) app, to administer ER questionnaires through tablets, and present the results of a study with 169 participants.

There is one main flaw in this article, in my opinion: the name chosen for the application, and consequently the title of the contribution, is misleading. I would expect an application named Multi-Modality Emotion Recognition (MMER) to actually perform automated Emotion Recognition from, for example, a multimedia flow containing both video and audio. On the contrary, the MMER app described in this work does not analyze any ER stimuli, but is used instead to administer single and multi-modal ER questionnaires. Therefore, I believe an alternative name should be chosen for the application and, consequently, the title should be rephrased.

Other than that, the work could be improved by looking at the following aspects:

The language used in this work could be made more fluent by rephrasing some paragraphs, e.g. the introduction.

The introduction section could be expanded and made more robust by adding some references to state-of-the-art techniques and tools that are currently used in Emotion Recognition, as well as challenges on the computational and technical sides, both for visual emotion recognition and prosody.

-It seems that no details are reported on the Emotional Speech Database in Taiwan dataset cited at row 129, from which speech segments were extracted. It is not clear to me which of the following options is true:
--the authors did not introduce the dataset previously. In that case, please briefly describe it.
--if the authors, instead, refer to the FAB (Chinese version) dataset, please make it clear. Moreover, if this is the case, why are prosody results reported in table 4 on 7 emotions while FAB contains only 5, according to table 2?

In section 2.2.2 please state how were the subtests designed. Were they administered in previous works? If not, were they designed by the authors, and according to which principles? What is the purpose of subtest 1?

Avoid expressions such as “it’s a pity”. Replace with “unfortunately”.

Row 61, place citation after authors.

Make tables less verbose, e.g. the amount of the items -> # items, choose within 7 emotions, replace first column with reference number only. Also, make definitions consistent across rows (e.g. images, color images, photographs, grayscale photographs. Reformulate the “test methods or procedures” entries, e.g.: “match each face with the word (Happiness, Disgust, Fear, Anger, Sadness, Surprise or Neutral)”-> 7 emotions classification.

In section 2.4, data analysis, please explain better the use of multi-dimensional models, how they were used and how they relate to the specific emotion recognition problem. Also, section 3.2.2 repeats the content of section 2.4; please delete or reformulate the similar content.

Figure 2 is too big and barely readable; please consider splitting it in subfigures, or replacing some parts with tables.

In the “discussion” section, avoid extensively repeating content available in other sections, such as “introduction” and “subtests”.

Reviewer 2 Report

In the paper “A Performance-based Measurement of Emotion Recognition: The Development of the Multi-Modality Emotion Recognition Mobile Application”.

The authors present MMER app as an application that measures an individual’s emotion recognition ability. However,

  • The authors should illustrate the main innovation of this research. Why was tablet selected for the task of interest? Why not a laptop or pc?. Why was used? It means there is program computer in any language.

  • Section 2 need an extreme makeover. As it is, it is not clear at all. For example, in line 112-113. “The tablet was used as an interface to calculate……”; Line 138-139 “We wrote the items for the app and used……”. What does is means?

  • I am sorry but the section “2.3 Measurement” should be explained with more details.

  • I think Figure 2 should be improved, is not clear.  

  • Finally, like any app, this must be replicable, however with the information provided this goal is not achieved. I suggest providing better structure and details.

Round 2

Reviewer 1 Report

The authors corrected all the issues reported in the first round.

Reviewer 2 Report

Dear authors,

My questions were answered and explained correctly. Although the title of the article was changed, in my opinion the new title is more appropriate to the content. For that reason, I have no objection for this paper to be published.